# Bending Performance and Reinforcement of Rocker Panel Components with Unidirectional Carbon Fiber Composite

**DOI:** 10.3390/ma12193164

**Published:** 2019-09-27

**Authors:** Huili Yu, Hui Zhao, Fangyuan Shi

**Affiliations:** 1School of Automotive Engineering, Chongqing University, Chongqing 400044, China; 2Crash Safety Department, Chongqing Changan Automobile Company, Ltd., Chongqing 401120, China; zhaohui@changan.com.cn (H.Z.); shify@changan.com.cn (F.S.); 3Crash Safety Department, State Key Laboratory of Vehicle NVH and Safety Technology, Chongqing 401120, China

**Keywords:** bending performance, rocker panel, unidirectional carbon fiber, simulation

## Abstract

Unidirectional carbon fiber composite material is one of the most common types of composites employed in vehicles, and its bending performance plays an important role in crash safety, especially in side pole impact. This study aimed to redesign one of the most important components of the side structure of a vehicle, the rocker panel, with unidirectional carbon fiber composite material. Our results show that it is not easy to acquire the same bending performance as that of a steel rocker panel by merely replacing it with carbon fiber material and increasing the wall thickness. Therefore, reinforcements were employed to improve the bending performance of the carbon fiber rocker panel, and a polypropylene reinforcement method achieved a weight reduction of 40.7% compared with high-strength steel.

## 1. Introduction

Carbon fiber is regarded as one of the most promising materials in the automotive industry due to its high strength and low density. Nowadays, it is utilized in the mass production of vehicles, especially for electric vehicles like the BMW i3 or NIO es6. Because it provides weight reduction while maintaining high strength of the body structure, it is possible to increase the driving mileage of electric vehicles significantly.

Although carbon fiber material has excellent mechanical characteristics, there are some obstacles in the way of its broad application, one of which is the simulation approach. It seems that no perfect material model exists that could characterize the strain rate effect and post-failure behavior of the material and simulate braided fabric composites. However, an appropriate material model can still possibly be chosen to simulate a given carbon fiber material and achieve the required accuracy.

Some scholars have studied different kinds of carbon fiber composite materials through different material models and different modeling approaches. Aleksandr Cherniaev [1] investigated three different LS-DYNA material models for modeling the axial crush of a CFRP energy absorber and identified the advantages and disadvantages of those models. XC. Sun [2] studied three different LS-DYNA finite element analysis (FEA) models in simulating low-velocity impact damage for two carbon fiber materials, and all the models provided reliable predictions. Post-failure behavior modeling is the key to composite damage prediction, so researchers like P. F. Liu et al. [3,4,5,6,7,8] explored different methods to predict the intraluminal and interlinear damage. In studying the carbon fiber component energy absorption performance, axial crushing is one of the most popular methods used by many experts [9,10,11,12,13,14,15,16]. Zheyi Zhang [17] studied the influence of geometry on the statistical energy analysis (SEA) of tubes in four different materials. The literature also shows that the modeling of composites has been applied in actual structures [18,19,20]. Ahad Torkestani [21] studied pedestrian head injury caused by engine hoods made of four different materials, and the results showed that the stacking sequence of composites had a great influence on the head injury criterion (HIC) value. Sha Yin [22] used a pyramidal lattice core to optimize the pedestrian protection performance of a composite engine hood and achieved a 25% weight reduction. 

Because of the mechanical performance of carbon fiber materials in the material direction, most investigations are focused on their energy absorption capability in axial crushing. However, for those components located in the occupant compartment zone, small deformations with large energy absorption would be preferred, which means that those components require good bending performance. Unidirectional carbon fiber composite material is one of the most common types of composites employed in vehicles, and its bending performance plays an important role in crash safety, especially in the case of pole impact. So, in this paper, unidirectional carbon fiber composite material was chosen to redesign the rocker panel, one of the most important components in the side structure of a vehicle. Its bending performance was studied and optimized.

## 2. Material Tests and Simulation Methods 

The unidirectional carbon fiber composite material used to replace the steel rocker panel was manufactured through the vacuum-assisted resin infusion method. The density of this composite is 1.53 × 10^−6^ kg/mm^3^. The carbon fiber was produced by Zhongfu Shenying Carbon Fiber Co., Ltd Suzhou, China) and contains 12 k carbon fiber in the axial direction. The matrix is an epoxy vinyl ester resin produced by Guangdong Broadwin Advanced Materials Co., Ltd (Guangdong, China). 

### 2.1. Coupon-Level Test

To determine the properties of this unidirectional carbon fiber composite material, material tests including of tension and compression at 0° and 90° in-plane shear were conducted according to ASTM (American Society of Testing Material) standards. The material properties are listed in Table 1.

The out-of-plane shear modulus, denoted GCA, was measured by three-point bending of a short beam flex sample [23,24]. The test specimen was 24 × 7 × 3.5 mm in size and consisted of 12 plies of 0° unidirectional layers. The average shear modulus calculated from the linear portion of the apparent shear stress–shear angle (Figure 1) was about 1.3 Gpa. The transverse shear modulus was not measured, and it was assumed to be equal to GCA.

### 2.2. Component-Level Test

The rocker panel of one electric vehicle was selected to be redesigned in the unidirectional carbon fiber composite material. The component section with a thickness different from that of the steel version is shown in Figure 2a. It was composed of two parts, a flat plate and a beam with a bowl profile section, which were connected by structural adhesive. Their stacking sequence was −45/45/0/0/0/45/−45 with a wall thickness of 2 mm. The whole component was 1200 mm in length.

The rocker panel component was subjected to a three-point bending test, which is an effective way to evaluate the bending performance. It was placed on two supports with 600 mm span and was crushed by an impactor moving at 2 mm/min. The top surfaces of the supports and impactor were cylindrical with diameters of 40 mm and 304.8 mm, respectively. The test diagram is shown in Figure 3a.

Tests were repeated three times, and the reacting force and displacement of the impactor were recorded. The results are shown in Figure 3b. It was found definitely that the force–displacement curve of each test could be divided into four stages. The first was the stage of elastic deformation. Here, the curves went up linearly before the impactor moved downward 5 mm. The second was the plasticity-like deformation stage. As the load increased, the reacting force continued to rise until it reached a peak. However, in this period, the curves went up nonlinearly and dispersedly. At the beginning of this period, local buckling on the top surface of the bowl profile beam was initiated in the contact area and spread to the side surfaces of the rocker panel component, which may be the reason for the nonlinear rise of the curve. Internal differences caused by the manufacturing method would be the main reason for the three scattered curves. The third was the failure stage. Although each test component reached a different peak force, they all experienced a sudden load drop at the same impactor displacement of around 20 mm when fracture was observed on the side surface of the bowl profile beam (Figure 3c). The last was the load keeping stage. The rocker panel component still carried a load of about 2–4 kN after passing the peak.

### 2.3. Simulation

The LS-DYNA MAT58 material model (*MAT_LAMINATED_COMPOSITE_FABRIC) was selected for modeling this unidirectional carbon fiber composite material. It assumes a nonlinear pre-peak and post-peak response, and the slope of the pre-and post-peak response can be changed via parameter ε_m_ (ε_m_ = E11T, E11C, E22T, E22C, GMS). ε_m_ is the corresponding strain at strength. This continuum damage mechanics model provided a smooth increase in the stress to strength and then a smooth decrease to the stress decided by scale factor n (n = SLIMT1, SLIMC1, SLIMT2, SLIMC2, and SLIMS). The typical stress–strain curve for this material model is schematically shown in Figure 4.

In plane stress conditions, the four failure criteria for composite materials can be simplified into the following forms:

For the tensile fiber mode, σ_11_ ≥ 0,
(1a)$$e^{2}={\left(\sigma_{11}/X_{t}\right)}^{2}-1\geq 0.$$

For the compressive fiber mode, σ_11_ < 0,
(1b)$$e^{2}={\left(\sigma_{11}/X_{c}\right)}^{2}-1\geq 0.$$

For the tensile matrix mode, σ_22_ ≥ 0,
(1c)$$e^{2}={\left(\sigma_{22}/Y_{t}\right)}^{2}+{\left(\tau /S_{c}\right)}^{2}-1\geq 0.$$

For the compressive matrix mode, σ_22_ < 0,
(1d)$$e^{2}={\left(\sigma_{22}/Y_{t}\right)}^{2}+{\left(\tau /S_{c}\right)}^{2}-1\geq 0.$$

A detailed description of this model can be found elsewhere [25].

The material parameters listed in Table 1 were adopted for the physical parameters of MAT58, and the strain rate effect was not considered according to the tensile test results (Figure 1) at different strain rates. Calibrations on the coupon level were done in the following. The nonphysical parameters of MAT58 were determined through the component-level calibration.

#### 2.3.1. Coupon-Level Calibration

The multilayered composite test samples were modeled by one-layer shell elements. The *PART_COMPOSITE keyword was used to model the composite and material, where the thickness and orientation properties of each ply could be defined separately. For a tensile test specimen consisting of 8 plies of 0° unidirectional layers, the thickness was about 2.4 mm. Each ply had the same material, same thickness (0.3 mm), and same fiber orientation (0°). However, the orientations of each element should be adjusted to match the reference direction of the layup through the Beta angle in *ELEMENT_SHELL_BETA. The element size of the finite element (FE) model for the coupon-level calibration was 3 mm. The Belytschko–Tsay element formulation was employed for the FE model. The stress–strain curves of coupon tests and simulations are shown in Figure 5a–e, showing that the FE model matched well with test results.

#### 2.3.2. Component-Level Calibration

The multilayered composite component was also modeled by one-layer shell elements, and each layer was modeled with the *PART_COMPOSITE keyword. The element size of the FE model for the component-level correlation was 5 mm, which was close to the mesh size of the vehicle crash simulation. The Belytschko–Tsay element formulation was employed. Adhesive was modeled by LS-DYNA MAT240 and was connected to parts by *CONTACT_TIE_NODES_TO_SURFACE. In order to save running time, the simulation loading velocity was 0.5 mm/ms, which was far beyond the testing velocity.

The results of the three-point bending from the simulation and tests on the component level are shown in Figure 6. It seemed impossible to match the test results without modifying ERODS, and the force–displacement curve of the simulation would rise after the impactor moved downward 4 mm. However, a quick improvement was achieved after ERODS was adjusted to 0.4. The maximum load and energy absorption obtained by the simulation and tests are compared in Table 2. The errors between those values were about 5.45% and 2.22%, respectively, which could meet the required precision for engineering applications.

## 3. Result

### 3.1. Comparison of Bending Performance

Safety performance has always acted as a contradictory element in lightweight design. Assessing the bending performance is an important means to evaluate the safety performance of this carbon fiber rocker panel. Therefore, its bending performance was compared with that of the steel version.

Two types of steel material commonly used for rocker panels were studied via the same simulation method as the carbon fiber rocker panel. One was high-strength steel (HSS), and the other was advanced high-strength steel (AHSS). The yield stresses of those two steel materials are 447 MPa and 1080 MPa, respectively, and their tensile stresses are 644 MPa and 1290 MPa, respectively. The thickness chosen for the two parts of the rocker panel component was 1.2 mm. A material card for the steel was calibrated through the coupon-level test, and the material parameters related to the strain rate were not defined.

The simulation results are shown in Figure 7. The reacting force of the steel rocker panel component was higher than that of the carbon fiber component from the beginning of deformation to the end. The peak reacting force values for carbon fiber, HSS, and AHSS were 6.82 kN, 10.54 kN, and 20.98 kN, respectively. The energy absorption values during crushing for those three materials were 218.69 J, 443.09 J, and 778.42 J, respectively. Therefore, unless the wall thickness is greatly increased, this kind of carbon fiber material cannot achieve bending performance as good as that of the steel material, although its tensile stress in the longitudinal direction was 1763 MPa.

### 3.2. Improvement of Bending Performance

Two approaches, foam core filling (Figure 8a) and polypropylene (PP) reinforcement (Figure 8b), were employed to improve the bending performance of the carbon fiber rocker panel component so as to not influence its exterior profile. The wall thickness of the PP reinforcement was 3 mm. The densities of those two materials are 5 × 10^−7^ kg/mm^3^ and 1.07 × 10^−6^ kg/mm^3^, and their elastic moduli are 1.6 GPa and 2.7 GPa, respectively.

The material models for modeling the foam core and PP reinforcement were LOW_DENSITY_FOAM and SAMP-1 in LS-DYNA. The foam material model was provided by Henkel, and the PP material parameters were identified by testing. The stress–strain curves of the PP material are presented in Figure 9.

The reinforced FE models experienced the same loading conditions as the carbon and steel component models. The simulation results are shown in Figure 10. Six parameters were acquired from the simulation results: component weight, maximum load, related displacement when reaching maximum load, energy absorption when the displacement of the impactor reached 50 mm, maximum load per unit weight of component, and energy absorption per unit weight of component. All of these results are shown in Figure 10c.

The results showed that the most effective way to increase the bending performance was the foam core filling method. It achieved not only the highest maximum load and energy absorption but also the highest maximum load per unit weight and energy absorption per unit weight. Although only 0.81 kg heavier than AHSS, the maximum load and energy absorption of the foam-core-filled carbon fiber rocker panel increased to 88.09 kN and 3107.76 J, around 4 times those of AHSS. 

The second-highest maximum load per unit weight and energy absorption per unit weight were achieved by the PP-reinforced carbon fiber rocker panel, second to the carbon fiber rocker panel. If weight reduction is the priority target, PP reinforcement is a considerably effective way to improve the bending performance when ignoring processing cost. The weight of the PP structure was 1.39 kg, and the maximum load and energy absorption increased by 143% and 132% compared with the carbon fiber rocker panel component, respectively. The bending performance of PP reinforcement was more excellent than that of HSS, but its weight was only 3.07 kg, a reduction by 40.7%. The performance of PP reinforcement could be improved if its rib arrangement and wall thickness were optimized.

## 4. Conclusion

In this paper, a unidirectional carbon fiber composite material was selected to be used in rocker panel components. The LS-DYNA MAT58 material model was employed in modeling this unidirectional carbon fiber composite. It showed good agreement in the coupon-level correlation, and a good component-level correlation was also achieved through further calibration on nonphysical parameters by trial and error.

Although its tensile stress in the longitudinal direction was 1763 MPa, the bending performance of the rocker panel component made of this kind of carbon fiber composite material could not be as good as that of a steel one unless the wall thickness were increased substantially and its profile changed.

One of the most effective ways found to increase the bending performance of composite for rocker panel components is the foam core filling method. It achieved not only the highest maximum load and energy absorption but also the highest maximum load and energy absorption per unit weight. PP reinforcement is also a considerably effective way to improve bending performance when ignoring the processing cost. Its bending performance could be superior to that of HSS, and the weight could be reduced by 40.7% compared to HSS.

## Figures and Tables

**Figure 1 materials-12-03164-f001:**
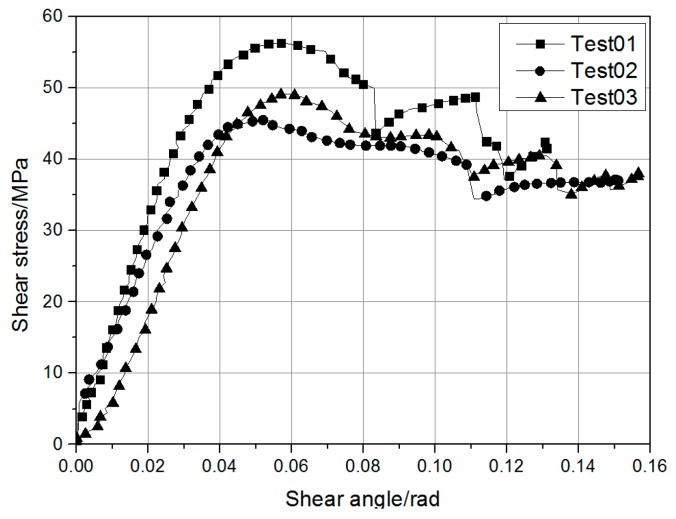
Shear stress vs. shear angle curve.

**Figure 2 materials-12-03164-f002:**
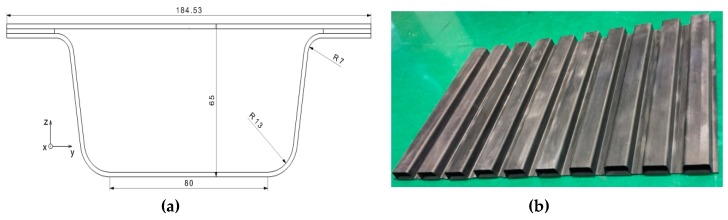
Rocker panel component in carbon fiber. (**a**) Cross section of the rocker panel. (**b**) Test specimens.

**Figure 3 materials-12-03164-f003:**
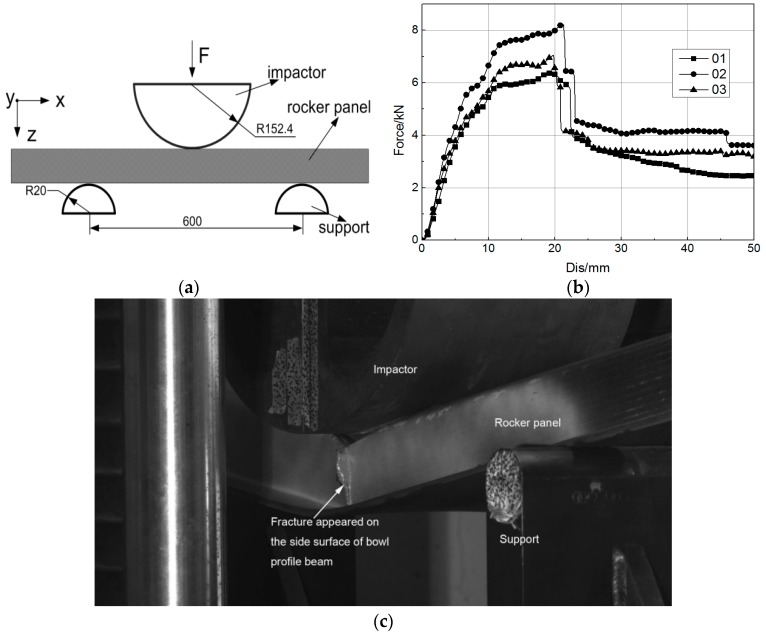
Three-point bending test. (**a**) Three-point test diagram. (**b**) Force–displacement curve. (**c**) Onset of fracture.

**Figure 4 materials-12-03164-f004:**
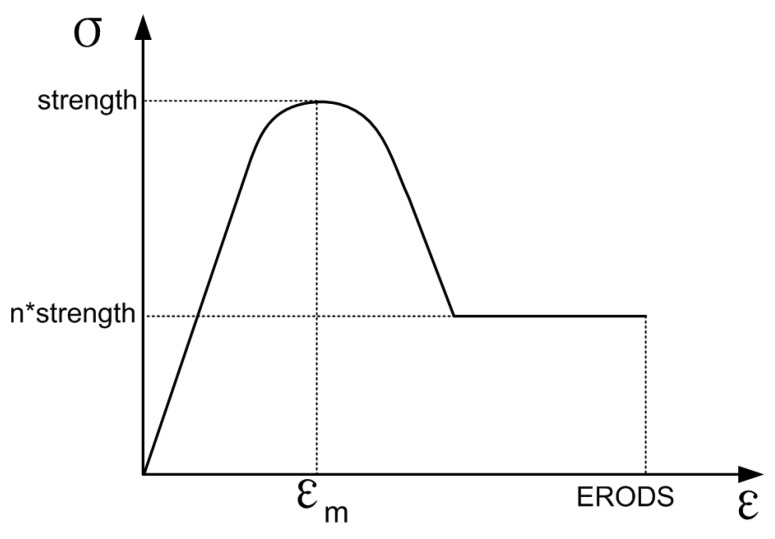
Shear stress vs. shear angle curve.

**Figure 5 materials-12-03164-f005:**
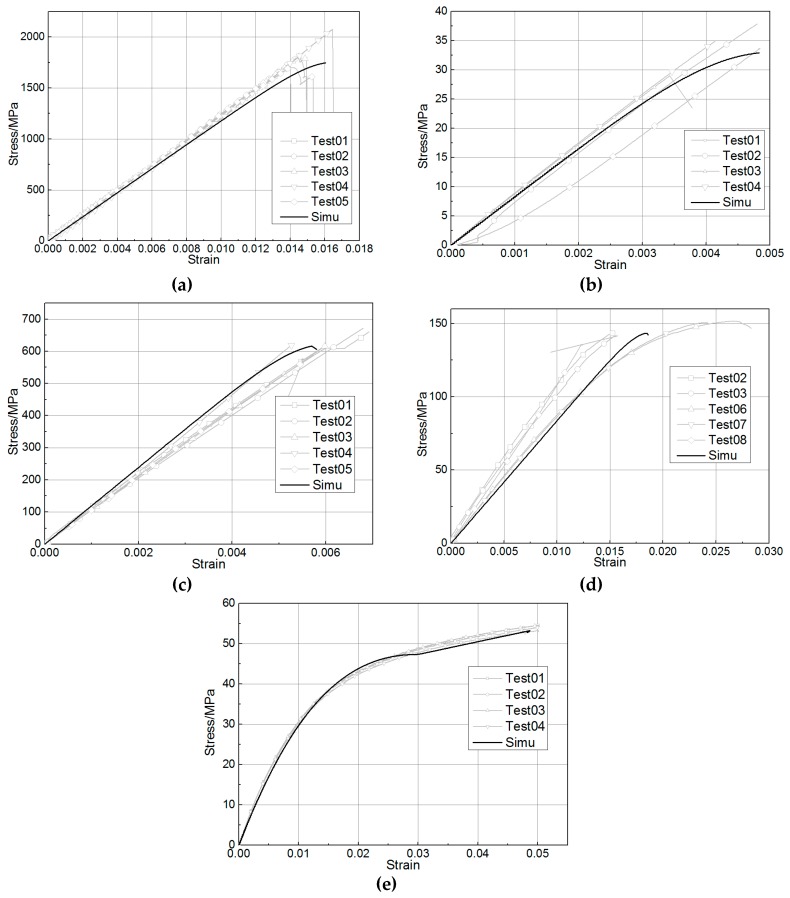
Comparison between test and simulation at the coupon level. (**a**) 0° tension. (**b**) 90° tension. (**c**) 0° compression. (**d**) 90° compression. (**e**) Shear.

**Figure 6 materials-12-03164-f006:**
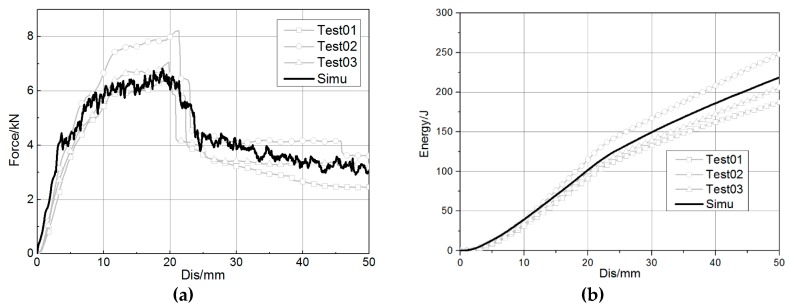
Comparison between test and simulation at the component level. (**a**) Force–displacement curve. (**b**) Energy absorption–displacement curve

**Figure 7 materials-12-03164-f007:**
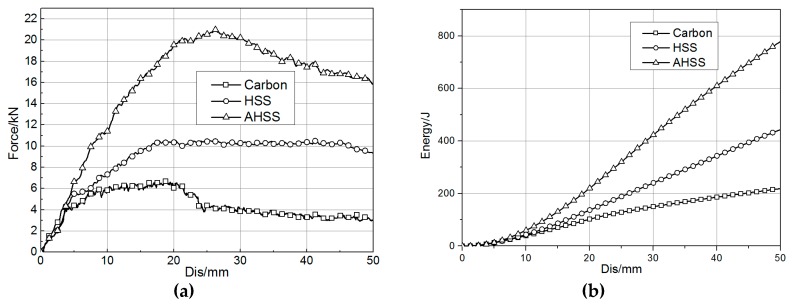
Three-point bending simulation results of steel and carbon fiber. (**a**) Force–displacement curve. (**b**) Energy absorption–displacement curve.

**Figure 8 materials-12-03164-f008:**
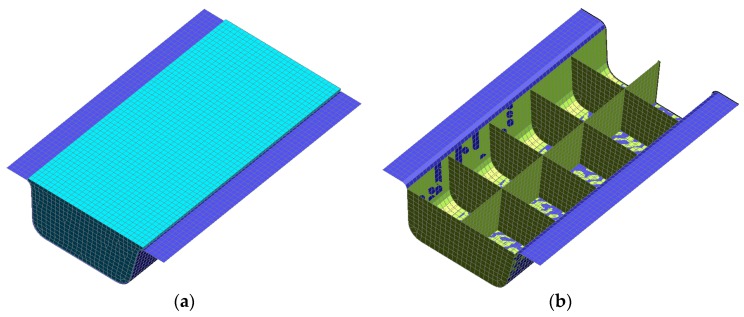
Reinforcement approaches. (**a**) Foam core filling. (**b**) Polypropylene reinforcement.

**Figure 9 materials-12-03164-f009:**
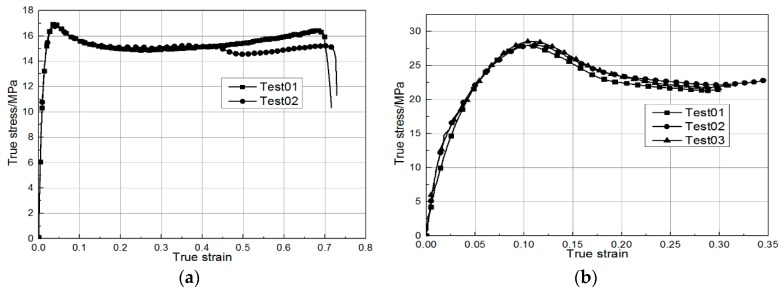
True stress–true strain curves of polypropylene material. (**a**) Tension. (**b**) Compression.

**Figure 10 materials-12-03164-f010:**
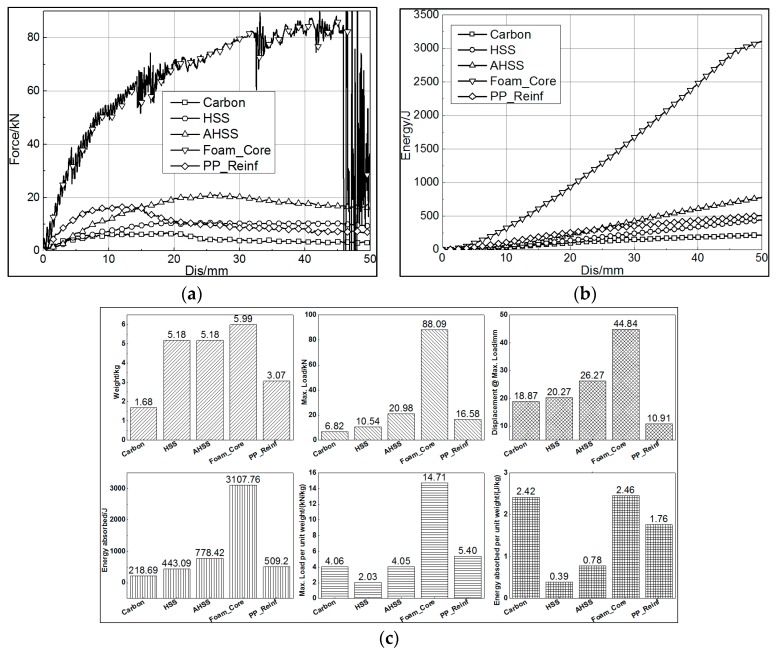
Comparison of the performance of different rocker panel components. (**a**) Force–displacement curve. (**b**) Energy absorption–displacement curve. (**c**) Histograms of parameters showing the bending performance of the rocker panels.

**Table 1 materials-12-03164-t001:** Material properties.

Property	LS-DYNA Parameter	Value
Modulus in longitudinal direction	EA	118.5 GPa
Modulus in transverse direction	EB	8.3 Gpa
Major Poisson’s ratio	PRBA	0.026
In-plane shear modulus	GAB	3.8 Gpa
Strain at longitudinal compressive strength	E11C	0.006
Strain at longitudinal tensile strength	E11T	0.014
Strain at transverse compressive strength	E22C	0.008
Strain at transverse tensile strength	E22T	0.005
Shear strength	GMS	0.05
Compressive strength in longitudinal direction	XC	0.619 Gpa
Tensile strength in longitudinal direction	XT	1.763 Gpa
Compressive strength in transverse direction	YC	0.144 Gpa
Tensile strength in transverse direction	YT	0.033 Gpa
Shear strength	SC	0.055 Gpa

**Table 2 materials-12-03164-t002:** Maximum load and energy absorption from the tests and simulation.

Terms	Max. Load/kN	Energy Absorption/J
Test01	6.37	187.8
Test02	8.22	248.22
Test03	7.05	205.78
Test_average	7.21	213.93
Simulation	6.82	218.69
Error	5.45%	2.22%

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
