# Peer review of "Bending Performance and Reinforcement of Rocker Panel Components with Unidirectional Carbon Fiber Composite"

_materials, 2019, doi:10.3390/ma12193164_

Round 1
Reviewer 1 Report
This study try to redesign a rocker panel with unidirectonal CFRP material. The study is well presented and structured, but some modifications should be done before publishing.
Please remark the novelty of the paper in the Abstract. Line 68: plead add the reference for the ASTM standard. Line 69: IN this sentence you mean you have carried out tension, compression and in plean shear tests, right? it is not clear if the shear tests are also done or not. You should mention how many repetition of each test have been done and please include in Table 1 the error or deviation of each calculated value. Include please, in section 2, the thickness of the material and the machined used in the tests. For the Figure 2a, please include in the text or in the caption the units. I guess all numbers are in "mm". Why in Figure 4 are different number of tests for each figure? Please expalin in the test, or used the same number of tests for all cases. Line 121: Are the units mm/ms correct?. Lines 138-139 Where did you get the value of the metal properties? If you have used any reference please include it. Curves from Figure 8 are taken from where? Did you tested the PP material also?. Please include some pictures from the simulations from the first part: Three point bending.Author Response
Please see the attachment.Thank you!

Reviewer 2 Report
The Authors use a commercial FEM software for studying the bending performance of automotive rocker panel made of unidirectional CFRP composite. The model is calibrated starting from the results of coupon tests on the material and from the results of three-point bending tests on the component. Then, some solutions for reinforcing the composite rocker panel are discussed, in order to reach the bending performance of a conventional high strength steel rocker panel.
The subject of the paper is interesting and have immediate applicative outcomes in the ongoing technological innovation process on structural components for the automotive industry. Anyway, in my opinion the paper presents some major weaknesses that have to be amended before publication. In the following, I list some comments that could be useful for revising the MS.
1) The title could be improved (“the reinforcement method”).
2) In the Introduction, the period between lines 27-32 gives the impression that the Authors are not completely confident with the huge amount of research results on the constitutive modeling of CFRP composite. I suggest to substantially improve it.
3) In Section 2.3, some considerations about the velocity of the test have to be included in order to justify the assumption of neglecting strain rate effects.
4) Special settings of the employed FEM software like “MAT58”, “*PART_COMPOSITE”, “MAT240”, “*CONTACT_TIE_NODES_TO_SURFACE”, etc., have to be conveniently explained.
5) In the discussion of the simulations, no information about boundary conditions, meshing strategy, finite elements employed, are given.
6) In the period between lines 123-129 it is not clear what is “ERODS” and if the diagram displayed in Figure 5(a) refers for the initial simulation or to the improved one (ERODS adjusted to 0.4). Also the oscillations observed in the diagram deserves a more in-deep discussion.
Reviewer 3 Report
N/A
Round 2
Reviewer 2 Report
Except for the change in the title, the Authors ignored all the suggestion I gave in the comments to the original MS. Thus, I think that also in the present version the paper presents some major weaknesses that have to be amended before publication. These weakness are pointed out in the comments 2) to 6) of my previous review report, reported below for convenience:
2) In the Introduction, the period between lines 27-32 gives the impression that the Authors are not completely confident with the huge amount of research results on the constitutive modeling of CFRP composite. I suggest to substantially improve it.
3) In Section 2.3, some considerations about the velocity of the test have to be included in order to justify the assumption of neglecting strain rate effects.
4) Special settings of the employed FEM software like “MAT58”, “*PART_COMPOSITE”, “MAT240”, “*CONTACT_TIE_NODES_TO_SURFACE”, etc., have to be conveniently explained.
5) In the discussion of the simulations, no information about boundary conditions, meshing strategy, finite elements employed, are given.
6) In the period between lines 123-129 it is not clear what is “ERODS” and if the diagram displayed in Figure 5(a) refers for the initial simulation or to the improved one (ERODS adjusted to 0.4). Also the oscillations observed in the diagram deserves a more in-deep discussion.
Author Response
The editor has given a number of professional and technical opinions, especially the impact of strain rate effects. We also considered this factor in the early experiment. The article has been modified according to the editorial opinions, and the specific contents are shown in the annex. Please see the attachment. Thank you!

Round 3
Reviewer 2 Report
The Authors introduced some of the modifications suggested.
I still have some concerns about the completeness of the description of the simulation strategy.
